# Comparing Wild and Cultivated *Arnica montana* L. from the Italian Alps to Explore the Possibility of Sustainable Production Using Local Seeds

**Valeria Leoni** [1] [iD]**, Gigliola Borgonovo** [2]**, Luca Giupponi** [1,3,*] [iD]**, Angela Bassoli** [2] [iD]**, Davide Pedrali** [1]**, Marco Zuccolo** [4]**, Alessia Rodari** [1] **and Annamaria Giorgi** [1,3]

[1] Centre of Applied Studies for the Sustainable Management and Protection of Mountain Areas (CRC Ge.S.Di.Mont.), University of Milan, Via Morino 8, 25048 Edolo, Italy; valeria.leoni@unimi.it (V.L.); davide.pedrali@unimi.it (D.P.); alessia.rodari@unimi.it (A.R.); anna.giorgi@unimi.it (A.G.)

[2] Department of Food, Environmental and Nutritional Sciences (DEFENS), University of Milan, Via Celoria 2, 20133 Milan, Italy; gigliola.borgonovo@unimi.it (G.B.); angela.bassoli@unimi.it (A.B.)

[3] Department of Agricultural and Environmental Sciences—Production, Landscape and Agroenergy (DISAA), University of Milan, Via Celoria 2, 20133 Milan, Italy

[4] Department of Medical Biotechnology and Translational Medicine, University of Milan, Via Vanvitelli 32, 20133 Milan, Italy; marco.zuccolo@unimi.it

\* Correspondence: luca.giupponi@unimi.it

**Abstract:** *Arnica montana* L. is an alpine herbaceous plant typical of nutrient-poor grasslands. It is a popular medicinal plant for the treatment of bruises, cuts and pain, and it is also an endangered alpine species. For this reason, the sustainable production of inflorescences instead of the spontaneous collection of plant material, coupled with the use of local ecotypes, should be incentivized. Inflorescences of a wild accession of arnica were compared versus an accession cultivated in Valsaviore (Italian Alps) in terms of seed germination performance and phytochemical characterization by high performance liquid chromatography (HPLC), nuclear magnetic resonance (NMR) and gas chromatography–mass spectrometry (GC-MS) techniques. The germination percentage was high (>75%) for both cultivated and wild seeds. The NMR spectra of arnica extracts were very similar and confirmed the presence of sesquiterpene compounds, esters of helenaline and dehydroelenaline. A significant high percentage of acetic acid methyl ester (38 µg/g) and the 2-methyl methyl ester of propanoic acid (31 µg/g) were found in cultivated arnica and were probably associated with fermentation processes linked to the traditional method of air drying on a trellis. The possibility of growing *A. montana* and a controlled local first transformation are important to incentivize local, good quality and sustainable production. The growing of seedlings "in loco" could be of great interest both for farmers and for natural conservation purposes.

**Keywords:** alpine medicinal plants; mountain agriculture; sesquiterpene lactones; VOCs

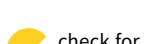

## 1. Introduction

The genus *Arnica* L. includes about 30 species that inhabit boreal and mountainous regions of the northern hemisphere [1]. Among them mountain arnica (*Arnica montana* L.) is an long-living herbaceous plant spread from South Norway and Latvia southwards to the Apennines and the south Carpathians, yet contained within the European continent [2]. For what concerns its habitat, *A. montana* was originally a common plant of nutrient-poor grasslands and dry heathlands [3] but habitat fragmentation and abandonment of traditional pastoralism practices have reduced its distribution [4]. *A. montana* is in fact locally present in mountain areas with low-intensity land use practices such as mountain-meadow farming [5], since it requires low nitrogen levels and the continuous absence of nitrogen fertilization [6], and regular cutting of the meadows after the flowering season.

Nowadays, the mountain meadows are rarely taken care of, and are gradually becoming shrublands and entering a natural succession process [7].

*A. montana* is also endangered by collection for herbal use. In Italy, a new law has been recently implemented in 2018 ("Testo unico in materia di coltivazione, raccolta e prima trasformazione delle piante officinali" D.lvo 21/05/2018 n, 75). Following the European Union (EU) Habitats Directive 92/43/EEC (Annex V), *A. montana* has been designated as a "plant species of community interest, whose exploitation may be managed and whose conservation should be encouraged". *A. montana* is included in the IUCN Red List of Threatened Species [8] and in the Red Data Books and Red Data Lists of many European countries [9].

*A. montana* is a traditional remedy and the main ingredient of numerous drug preparations, having a well-known antiseptic, antifungal, antimicrobial, and antibiotic activities [10]. Arnica tincture as described in the European Pharmacopoeia has been approved by the ESCOP for the external treatment of different ailments related to injuries and accidents [11], linked especially to sport activities, and it is described as a topic treatment for skin bruises, irritations, contusions, and pain, presenting antiseptic, antiphlogistic, analgesic and anti-inflammatory properties [12]. The anti-inflammatory effects of arnica flowerheads are mainly due to some of its secondary metabolites, the sesquiterpene lactones (SLs) of the $10\alpha$-methylpseudoguaianolide type, which have been shown to inhibit a variety of inflammatory mediators [13]. Sesquiterpene lactones (e.g., helenalin, dihydrohelenalin and their esters) have important biological activities analogous to those of corticoid steroids [14]. The contents of helenalin, $11\alpha$,13-dihydrohelenalin and their ester derivatives [11] are thus the parameters that define the quality of arnica, although, a better activity is observed for total plant extracts compared to pure compounds isolated from arnica flowers [15].

For its important role as medicinal plant and the depletion of its habitat, conservation protection of this species has been introduced in several European countries besides Italy [16]. Cultivation of *A. montana* could supply the herbal demand and help to preserve the natural populations. To date this could be possible, but it is not easy and rather costly [17]. In 1990, Bomme (LFL) produced the first cultivar of *A. montana*, named 'Arbo' [18], which became a commercial cultivar a few years later. The 'Arbo' cultivar derives from numerous accessions collected in the wild in Austria, Germany, and Switzerland and from different botanical gardens in these countries. Free pollination between 55 different accessions was possible during the breeding process [19] so it is not possible to assign the relative contribution of the individual accessions to the genome of *A. montana* cv. "Arbo", that is essentially the only commercially recognized arnica cultivar, even if the selection followed the process seen above (slightly different from the usual genetic improvement). The cultivated material entered the arnica extracts and ointments market a decade ago and most of the companies producing official plant extracts would prefer to use such a material, but it is still too expensive. To manage arnica and other natural resources in an effective and sustainable way research is fundamental, not only concerning arnica distribution, habitat and ecology, but also considering drying methods, arnica trade and the socioeconomic context [17]. Both the quality of the collected material and the processing methods define the quality of dried arnica flower-heads. Drying is one of the most essential processing steps and it is considered by the Italian law a first transformation process that can be done on-farm and it can be then established locally. Local drying could contribute to create local added value, as dried arnica can be sold at higher prices than the fresh material and it is a higher quality product, since the raw material is immediately processed to better preserve the drug. Temperature, humidity, and other factors can influence the drying process and appropriate drying protocols must be formulated locally as conditions can vary depending on the environment of each place [17].

Very often essential oils were studied to investigate the volatile composition of different botanical materials, but very little research was realized on the dried material, that is the starting point for most of the production of arnica ethanol tincture as regulated

by European Pharmacopoeia (communication EMA/HMPC/198793/2012 Committee on Herbal Medicinal Products (HMPC)). For the study of the phytochemical profile of arnica and its active compounds, solid phase microextraction (SPME) is a simple and fast modern tool used to characterize the volatile fractions of medicinal plants [20] and foods [21] and offers a valid alternative to hydrodistillation (HD) for gas chromatographic analysis of essential oils from different sources [22]. SPME coupled with GC-MS can avoid the losses and degradation of volatile constituents that happen with HD, very often used for the characterization of officinal plants volatiles [23,24] and eliminates most of the drawbacks of extracting organics, including high cost and excessive preparation time. In SPME, the analytes are extracted from fluid or solid matrices by headspace (HS) or direct immersion sampling (DI). The application of $^1$H-NMR techniques for the determination of the amount of biologically active compounds in plant extracts [25,26] has also increased in the past years. The main advantages of the $^1$H-NMR method are its non-destructive nature, no need for chromatographic purification of individual compounds, no need of authentic samples for plotting calibration curves and the short duration of the time needed for measurements. NMR analysis has been already applied for qualitative characterization of the lactone components of arnica extract [27].

Based on these considerations, the aims of this study were to test the germination performance of a wild accession of arnica versus a cultivated one and the phytochemical characterization of wild and cultivated arnica dried inflorescences to explore if the wild germplasm is eligible for arnica flower-head production both for its chemical composition and germination performances. Locally dried inflorescences were sampled and studied as the quality of dried arnica flower-heads depends on both the quality of the collected material and the processing methods [17]. With this information, it is hoped to support the local cultivation and processing of arnica and investigate if it is possible to use wild germplasm. The latter aspect could be useful, in fact, for conservation purposes other than incentivizing arnica cultivation instead of the collection of wild material.

## 2. Materials and Methods

### 2.1. Plant Material

Valsaviore is an alpine valley on the orographic left of the upper-middle Valle Camonica (Italy; Latitude: 46°07′30″ N; Longitude: 10°29′45″ E). This area is included in the Temperate Oceanic bioclimate [28] and within the Northeastern Alps Ecoregional Subsection (Central and Eastern Alps Section, Alpine Province, Temperate Division) according to [29].

The seeds and inflorescences (flos) of wild arnica were collected in a 50 km$^2$ area of Adamello mountain area in Valsaviore, from 1100 to 1300 m a.s.l. The flos of cultivated arnica were obtained from the mountain farm Shanty Mae located in in the municipality of Saviore dell'Adamello (latitude 46°4′53″04 N, Longitude 10°24′2″52 E, elevation 1100 m a.s.l.), always in Valsaviore. The cultivated seeds used for the test were purchased from the nursery providing the seedlings to Shanty Mae farm, located in Valle d'Aosta. The seeds of the nursery come from different accessions collected in the wild in the Western Alps mixed with the "Arbo" cultivar. As happened for the latter, free pollination between different accessions was possible during the breeding. The farm renews its crops every three years with seedlings from the nursery. The flower-heads used for the study where the ones from the first year of production of the seedlings. Arnica fields were prepared by ploughing, manual removal of weed roots and successive mechanical tilling. Nor fertilization nor irrigation were performed.

Arnica was in both cases (cultivated and wild) collected at full flowering when the ligulate florets opened, as different studies demonstrated being the balsamic time for flos of *A. montana* [30] and this is also confirmed by traditional use of *A. montana*. Wild flos were immediately transported and oven dried in the laboratory at 25–30 °C for one week. Cultivated arnica was dried at Shanty Mae farm using the traditional method of air drying on a trellis protected from light and the leftover material was oven dried when

the trellises were full. Then, a random sample was brought to the laboratory for analysis. The flos samples were well-preserved until analysis following the recommendation in [31]. The seeds were air dried to avoid damage or rotting during cold storage, cleaned, and stored in darkness at 4 °C for six months (cold stratification), after which germination tests were performed.

### 2.2. Germination Trials

Cultivated seeds and wild seeds collected in Adamello Park were sterilised in 15% sodium hypochlorite (NaClO) solution for five minutes and then rinsed with distilled water. Immediately afterwards the seeds were transferred to sterile Petri dishes. Three filter paper discs (Whatman No. 3) were placed in each Petri dish with 25 seeds placed on top. Each Petri dish contained a single type of seeds (cultivated seed and seed collected in Adamello Park) and underwent one of the following treatments:

- addition of 5 mL of distilled water (control);
- addition of 5 mL of a solution containing 50 mg L$^{-1}$ of gibberellic acid (GA3);
- addition of 5 mL of a solution containing 100 mg L$^{-1}$ of GA3;
- addition of 5 mL of a solution containing 200 mg L$^{-1}$ of GA3.

Petri dishes were hermetically sealed with parafilm to prevent evaporation. Four replicas were performed for each treatment. Seeds were incubated for 30 days in a germination chamber (FDM—Series C, Fratelli della Marca, Rome, Italy) in the following environmental conditions: 12/12 h light/dark cycle at 23/10 °C. Every two days the Petri dishes were re-randomized and seeds showing radicle and cotyledons emergence were recorded as germinated and removed from the plates.

At the end of the germination tests the germination percentage (GRP) and germination speed coefficient (GSP) were calculated according to [32]. In detail, GRP was calculated as follows:

$$GRP = \left( \frac{\sum_{i=1}^{k} n_i}{N} \right) * 100 \tag{1}$$

where $n_i$ is the number of germinated seeds in the $i$ time and $N$ is the total number of seeds in each experimental unit. GSP was calculated using the following formula:

$$GSP = \left( \frac{\sum_{i=1}^{k} G_i}{\sum_{i=1}^{k} G_i X_i} \right) * 100 \tag{2}$$

where $G_i$ is the number of seeds germinated in the $i$ time and $X_i$ is the number of days from sowing. Both GRP and GPS are expressed in percentages.

For each plant material (cultivated seed and wild seed) a one-way ANOVA test was used to evaluate the effect of the treatments on GRP and GSP. When significant ($p < 0.05$) effects existed, differences were tested by Tukey's post-hoc test. Finally, differences between two type of seeds for GRP and GSP were determined using Student's *t*-test from R 3.6.3 statistical software [33]. *p*-value less than 0.05 was considered statistically significant. GRP, GSP and ANOVA were calculated/performed using the "GerminaR" package [32] of R.

### 2.3. Phytochemical Analysis

2.3.1. Chemicals and Reagents

Chloroform-d (99.8%), 4-methyl-2-pentanone (internal standard), chloroform, hexane, diethyl ether, methanol and 2-propanol (HPLC grade) were purchased from Sigma Aldrich (Milan, Italy). SPE cartridges (GracePure™ SPE C18-Max) were purchased from SepaChrom (Rho MI, Italy).

2.3.2. Preparation and Analysis through HPLC and NMR of Crude Lactone Extracts

Dried flowers of *A. montana* (10.00 g) were shredded and extracted twice with chloroform in an ultrasonic bath at room temperature for 10 min following the literature

protocol reported in [34]. After evaporation of the solvent under reduced pressure two crude extracts, were obtained as yellow viscous oils, (yields: 3.9 ± 0.9% for wild arnica and 4.6 ± 0.4% for cultivated arnica). An Euronic 22 ultrasonic bath (Argo Lab, Carpi, Italy) operating at 50 KHz and 50 W power was used. A quantity of about 10 mg of each extract was solubilized in 1 mL of methanol and passed through a SPE cartridge previously conditioned with 3 mL of a mixture methanol/water 3:2, and then eluted with 1 mL of the same mixture. The eluate was filtered through a 0.45 nylon filter and analyzed in HPLC and by NMR. The analytical equipment used consist of a ProStar HPLC instrument (Varian Analytical HPLC—Agilent, Stevens Creek Blvd, Santa Clara, CA 95051, USA) equipped with a ternary pump and a UV-Vis detector and managed by the Chem software Open Lab (Agilent, Stevens Creek Blvd, Santa Clara, CA 95051, USA). An $RP_{18}$ column (Lichrospher 250 × 4 mm, 5 µm, Phenomenex, 411 Madrid Avenue, Torrance, CA 90501-1430, USA) was used under gradient conditions with a mixture of methanol/water, flow 1 mL/min, detection at 225 nm. A linear gradient from 55:45 to 40:60 water/methanol over 35 min was used. [16]. NMR spectra were recorded on an Advance 600 MHz spectrometer (Bruker Scientific LLC 40 Manning Road Billerica, MA 01821, USA) equipped with 5 mm probe. The crude extract was further separated into four fractions by column chromatography on silica gel using as eluent a mixture of hexane:diethyl ether 7:3. Each fraction was analyzed by NMR spectroscopy to determine the qualitative lactone composition and the presence of specific signals of this group of bioactive compounds. The detection of the signals for each individual compound was performed using a combination of 1D and 2D NMR techniques ($^{13}C$, COSY, HSQC and HMQC spectra). $^{1}H$- and $^{13}C$-NMR chemical shift values determined in our study for helenalin and 11,13-dihydrohelenalin skeleton types as well as for ester residues corresponded to literature data for fraction f2 and f4. Fraction f2 and f4 were then identified as the ones containing the sesquiterpene lactones that were found in a percentage of 0.48 ± 0.07%, comparable with [16], where a range among 0.31–1.01% is specified.

### 2.3.3. Volatiles Analysis through SPME GC-MS

Samples (1.0 g each) were ground in a high intensity planetary mill to obtain superfine wild and cultivated arnica inflorescence powder. The mill was vibrating at a frequency of 25 Hz for 1 min, using two 50 mL jars with 20 mm stainless steel balls. Prior to use, jars were precooled with liquid nitrogen. These analytical procedures were described in detail in some reported research for the characterization of cannabis inflorescences [31,35,36]. In brief, inflorescence powder (100 mg) previously grinded was weighed and put into 20 mL glass vials along with 100 µL of the IS (4-methyl-2-pentanone, 20 mg/mL in 2-propanol). Each vial was fitted with a cap equipped with a silicon/PTFE septum (Supelco, Bellefonte, PA, USA) and passed in an ultrasonic bath for 10 s at 30 °C. To keep the temperature constant during analysis (37 °C), the vials were maintained in a cooling block (CTC Analytics, Zwingen, Switzerland). At the end of the sample equilibration time (30 min), a conditioned (60 min at 280 °C) SPME fiber was exposed to the headspace of the sample for 120 min using a CombiPAL system injector autosampler (CTC Analytics). Analyses were performed with a Trace GC Ultra coupled to a Trace DSQII quadrupole mass spectrometer (MS) (Thermo-Fisher Scientific, Waltham, MA, USA) equipped with an Rtx-Wax column (30 m × 0.25 mm i.d. × 0.25 µm film thickness) (Restek, Bellefonte, PA, USA). The oven temperature program was from 35 °C, held for 8 min, to 60 °C at 4 °C/min, then from 60 to 160 °C at 6 °C/min and finally from 160 to 200 at 20 °C/min. Helium was the carrier gas, at a flow rate of 1 mL/min. Carry over and peaks originating from the fibers were regularly assessed by running blank samples. After each analysis fibers were immediately thermally desorbed in the GC injector for 5 min at 250 °C to prevent contamination. The MS was operated in electron impact (EI) ionization mode at 70 eV. A $C_8$-$C_{22}$ alkanes mixture (R 8769, Sigma, Saint Louis, MO, USA) was run under the same chromatographic conditions as the samples to calculate the Kovats Retention Indices (RI) of the detected compounds [37–41]. The mass spectra were obtained by using a mass selective

detector, a multiplier voltage of 1456 V, and by collecting the data at a rate of 1 scan/s over the m/z range of 35–350. Compounds were identified by comparing the Kovats retention indices with the literature data and through the National Institute of Standards and Technology (NIST, Gaithersburg, ML, USA) MS spectral database. The quantitative evaluation was performed using the internal standard procedure and the results were finally expressed as μg/g. For both chemotypes, all analyses were done in five biological replicates and three technical replicates.

Differences between the two type of plant material (wild and cultivated arnica inflorescence) for the quantitative analysis were determined using Student's *t*-test from the R software [33]. A *p*-value of less than 0.05 was considered statistically significant. VOCs that resulted significant at the t-test were employed in the multidimensional scaling (MDS) to highlight the most important differences between the two type of plant material. MDS shows measurements of similarity (or dissimilarity) among pairs of objects as distances between points of a low-dimensional multidimensional space. MDS was carried out considering the Euclidean distance and was performed using R software [33].

## 3. Results

### 3.1. Seeds Germination

Figure 1 shows the results of the germination tests. The graphs of the germination curves of the two types of seeds (wild and cultivated) are similar and show that the maximum germinability is reached after 10–12 days from sowing. The GRP values are always higher than 75% and the treatments have no significant differences on GRP. The GSP values are higher for cultivated seeds and the Tukey's test revealed significant differences only between the treatments related to wild seeds. There is a significant difference between the GSP values of the wild seeds treated with water (which have lowest GSP) and those of the seeds treated with the solution with highest concentration of GA3 (which have highest GSP). GRP and GSP comparisons between wild and cultivated seeds of arnica show that GSP is significantly higher for cultivated seeds in each treatment (Figure 2). Only the treatments with 100 mg $L^{-1}$ and 200 mg $L^{-1}$ GA3 solution had significantly different effects on GRP which, in the first case, is higher for wild seeds while, in the second case, it is higher for cultivated seeds.

### 3.2. Phytochemical Features

The HPLC chromatographic profiles of the two arnica extracts were complicated due to the presence of several peaks but very similar for what concerns the qualitative composition. In particular some peaks occur in the two samples with little difference in the peak's intensity (Figure 3). Details of the $^1$H-NMR spectra of arnica extracts demonstrate an abundance of signals (Figure 4) and confirm the presence of sesquiterpene compounds, esters of helenaline (H) and dehydroelenaline (DH). The NMR profiles of the two arnica samples are very similar and present a similar qualitative composition, as can be seen in Figure 4. Typical proton resonances of helanolides were easily assigned on the basis of their proton and carbon chemical shifts by comparison with data reported in the literature [34]. The signals at 6.10 and 7.95 ppm were assigned to the olefin protons of the helanolides ketofuran ring, while the proton signals of the double bond present in position 13 of the helanolides structures (HT, HM, HIB, HMB) were found in the spectra with resonances at about 6.40 ppm as broad signals with a coupling constant value of 2.8 Hz with a signal between 6.08–6.11 ppm that appears as a multiplet. Carbon resonates at 125 ppm. In the NMR spectrum the dihydrohelanolides structures (DHT, DHM, DHIB and DHMB) an extra methyl group appearing a doublet signals between 1.45–1.48 ppm with a *J* value of 7.50 Hz is observed. The other two methyl groups, also present in the helanolides, resonate specifically between 0.96 and 0.98 ppm as a singlet for CH$_3$-15 and between 1.15 and 1.19 as a doublet with *J* = 6.72 Hz for CH$_3$-14. The carbon resonances for the methyl groups in positions 13–15 were located at 10.85–10.92; 19.68–19.71 and 17.52–17.53 ppm, respectively. The main compounds identified were thus 6-O-(2-methylbutyryl)-helenalin (HMB) present

in fraction four (f4) in a mixture with 6-O-isobutyryl-11α,13-dihydrohelenalin (DHIB), 6-O-(2-methylbutyryl)-helenalin (HIB) and trace amounts of 6-O-(2-methylbutyryl)-11α,13-dihydrohelenalin (DHMB) (Figure 5). The second fraction (f2) consists of a mixture of 6-O-(2-methylbutyryl)-helenalin (HMB), 6-O-(2-methylbutyryl)-helenalin (HIB) and 6-O-metthacryloylhelenalin (HM), while the first and the third fractions (f1 and f3) did not contain any sesquiterpene lactones. As we can see in Table 1, cultivated arnica resulted significantly richer is HM, HIB and HMB, while the wild genotype resulted extremely significantly richer in some unidentified compounds.

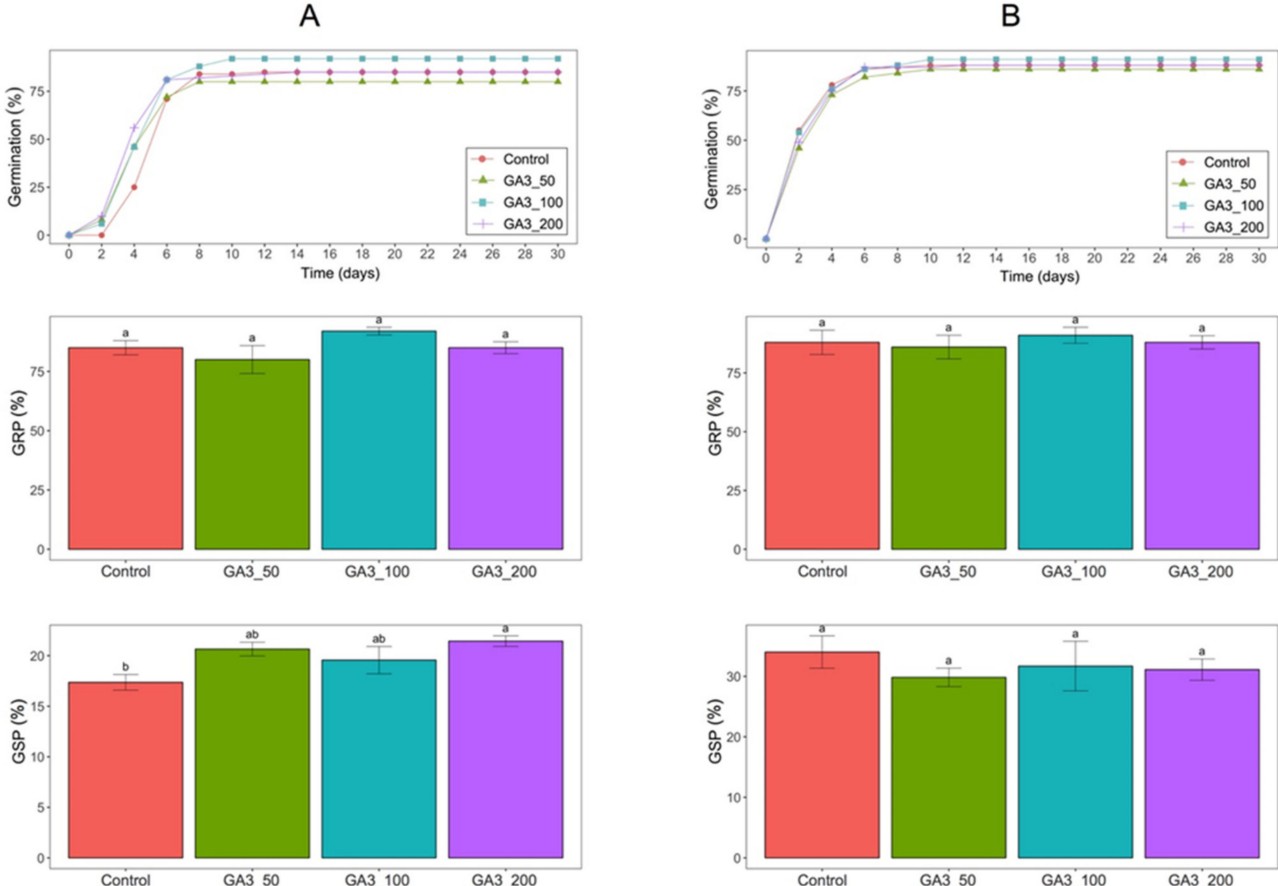

**Figure 1.** Germination line, germination percentage (GRP) and germination speed coefficient (GSP) for each treatment of the two types of *A. montana* seeds: wild (**A**) and cultivated (**B**). Key of treatments: Control, water; GA3_50, 50 mg L$^{-1}$ solution of GA3; GA3_100, 100 mg L$^{-1}$ solution of GA3; GA3_200, 200 mg L$^{-1}$ solution of GA3. Different letters above bars indicate significant differences ($p < 0.05$) among treatments.

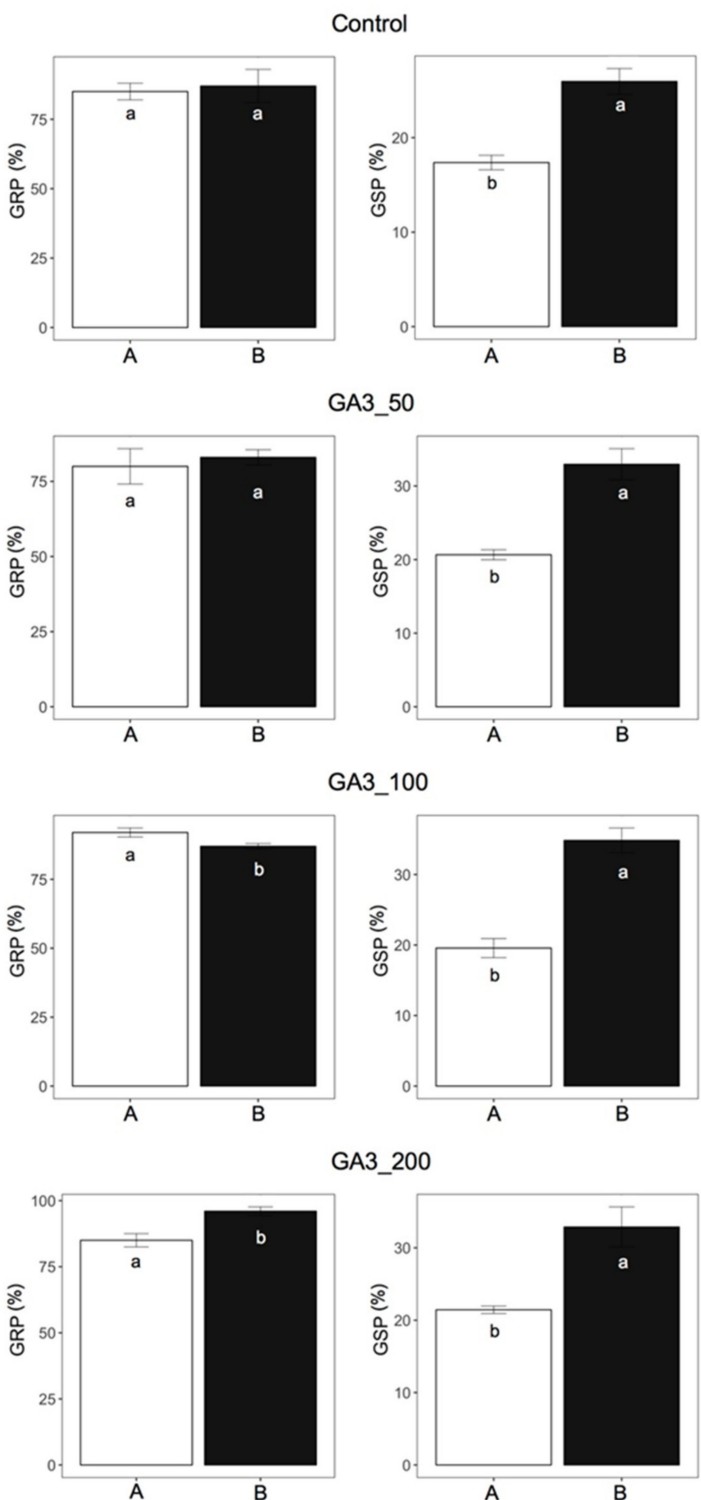

**Figure 2.** GRP and GSP comparison of wild (A) and cultivated (B) seeds of *A. montana* for each treatment: Control, water; GA3_50, 50 mg L$^{-1}$ solution of GA3; GA3_100, 100 mg L$^{-1}$ solution of GA3; GA3_200, 200 mg L$^{-1}$ solution of GA3. Different letters in the bars indicate significant differences ($p < 0.05$) among treatments returned by *t*-test.

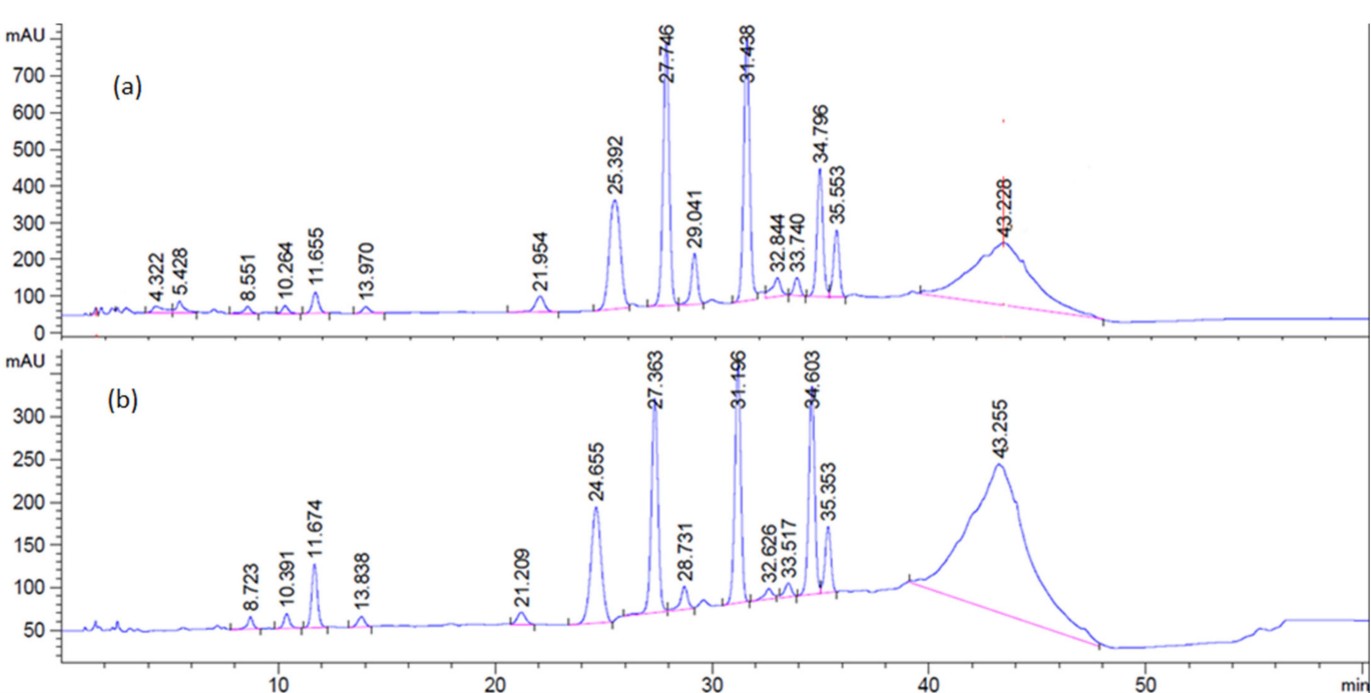

**Figure 3.** Comparison of HPLC profile of the crude lactones fraction: (**a**) wild arnica; (**b**) cultivated arnica.

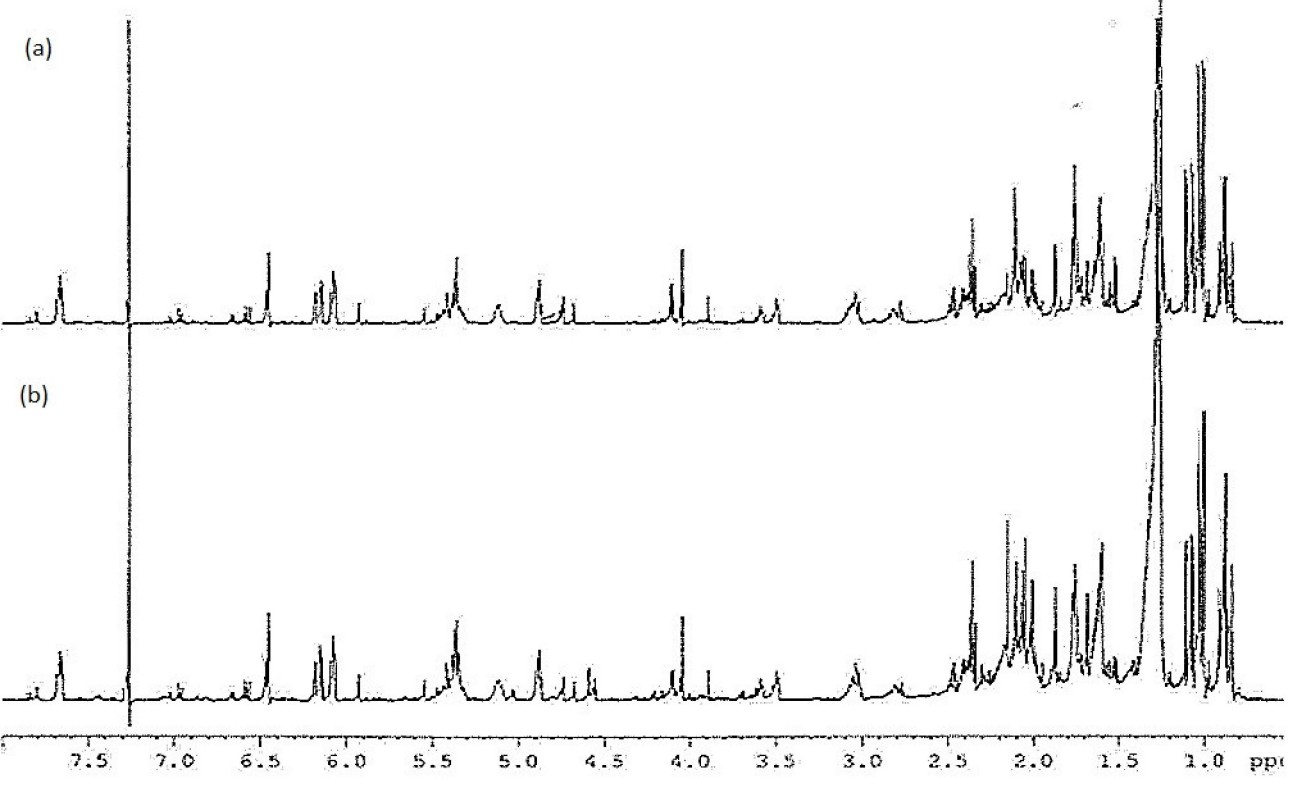

**Figure 4.** Comparison of 1H NMR spectrum of the crude lactones fraction: (**a**) wild arnica; (**b**) cultivated arnica.

**Figure 5.** Structures of sesquiterpene lactones found in *Arnica montana* L.

**Table 1.** sesquiterpene lactones content measured by $^1$H NMR and HPLC.

| RT [a] | Compound | Wild Arnica | | Cultivated Arnica | | p-Value | Signif. Code |
|---|---|---|---|---|---|---|---|
| | | Mean [b] | ±SD [c] | Mean [b] | ±SD [c] | | |
| 25.39 | peak 1 | 13.41 | 1.08 | 23.34 | 2.46 | <0.0001 | *** |
| 27.74 | peak 2 (HM) | 17.50 | 1.11 | 19.35 | 2.66 | 0.1893 | ns |
| 29.04 | peak 3 (n.i.) | 3.28 | 0.24 | 2.65 | 0.23 | 0.0029 | ** |
| 31.44 | peak 4 (n.i.) | 17.21 | 1.31 | 22.60 | 1.65 | 0.0005 | *** |
| 34.79 | peak 5 (HMB) | 7.44 | 1.06 | 16.24 | 2.38 | <0.0001 | *** |
| 35.55 | peak 6 (HIB) | 3.91 | 0.37 | 5.47 | 1.60 | 0.0649 | ns |
| 43.23 | peak 7 (n.i.) | 37.26 | 4.47 | 12.00 | 7.61 | 0.0002 | *** |

RT [a]: retention time (min); Mean [b]: Mean value ($n = 5$); Data are expressed in percentage; SD [c]: Standard deviation ($n = 5$); Signif. Code[d]: **, $p < 0.01$; ***, $p < 0.001$ ns, not significant.

For what concerns the volatiles fraction, wild and cultivated arnica all contained almost the same compounds (except for methyl valerate that was present only in the cultivated arnica) and differed only quantitatively for 44 out of 71 compounds identified. The terpenes and terpenoids fractions were higher in wild arnica (210 µg/g, the 65% of total volatiles, instead of 173 µg/g for cultivated arnica, the 50% of total volatiles) out of a total content of 321 µg/g of total volatiles for wild arnica and 348 µg/g for cultivated arnica. The major compounds in *A. montana* volatiles composition were found to be germacrene D (found in a quantity of 26 µg/g in wild arnica and 10 µg/g in the cultivated one), α-bergamotene (18 µg/g in the wild cultivar and 11 µg/g in the cultivated flos), cymene (14 µg/g) limonene (11 µg/g) and α-phellandrene (15 µg/g) in the cultivated arnica and δ-cadinene (13 µg/g) in the wild one. In the cultivated inflorescences, very significantly higher quantities of linear and branched hydrocarbons such as acetic acid methyl ester, propanoic acid 2-methyl methyl ester, caprylic acid methyl ester, acetic acid

and 2-methyl propanoic acid were found. Other compounds found very significantly higher amounts in the cultivated arnica flos were α-phellandrene, β-sesquiphellandrene, limonene, cymene, and borneol butyrate (while bornyl acetate was higher in wild arnica). The wild arnica, on the other hand, contained very a high quantity of hexanal (while 2-hexenal was found higher in cultivated arnica flos) more than three times than the cultivated flos, other than a higher quantity of α-bergamotene, more than double the quantity of germacrene D and three times the quantity of δ-cadinene and Y-muurolene. Many other compounds identified as sesquiterpenes were found in higher quantity in the wild arnica (sesquiterpene_1; sesquiterpene_2; sesquiterpene_4; sesquiterpene_5; sesquiterpene_6; sesquiterpene_7; sesquiterpene_8; sesquiterpene_9; sesquiterpene_11; sesquiterpene_13; sesquiterpene_14; sesquiterpene_15).A significantly high percentage of acetic acid methyl ester (38 µg/g) and 2-methyl methyl ester of propanoic acid (31 µg/g) were found for cultivated arnica.

The multidimensional scaling (MDS) undelining these differences is shown in Figure 6, where at the top of the graph (encircled in blue) one can see the exclusive or most abundant compounds in wild arnica while the exclusive or most abundant compounds in cultivated arnica are found in the lower part of the graph (surrounded in red).

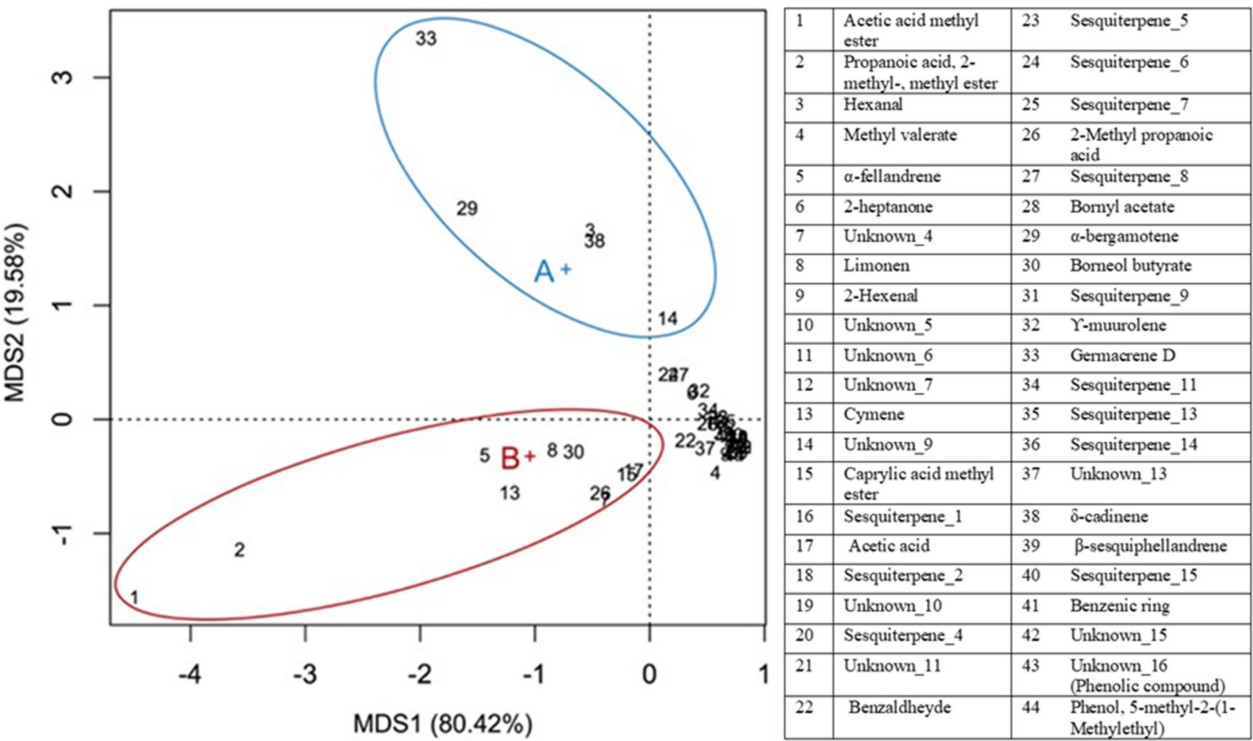

**Figure 6.** MDS biplot of VOCs in a significatively different quantity in wild and cultivated arnica. The blue cross and the red cross indicate wild (A) and cultivated arnica (B) respectively. The compounds that characterize wild arnica and those that characterize commercial arnica are highlighted with blue and red ellipses, respectively. Numeric codes refer to the compound shown in the table.

The results of SPME/GC-MS analysis are showed in Table 2.

**Table 2.** Volatile compounds identified in wild (A) and cultivated (B) arnica collected in Valsaviore.

| RT [a] | Compounds | Wild Arnica | | Cultivated Arnica | | | |
|---|---|---|---|---|---|---|---|
| | | Mean [b] | ±SD [c] | Mean [b] | ±SD [c] | *p*-Value | Signif. Code |
| 2.33 | Acetic acid methyl ester | 11.56 | 1.90 | 38.73 | 4.43 | 0.0006 | ** |
| 2.93 | Unknown_1 | 7.99 | 7.20 | 3.99 | 1.78 | 0.9876 | ns |
| 3.52 | 2-Methyl-propanoic acid 2-methyl-, methyl ester | 10.51 | 4.14 | 31.57 | 9.36 | 0.0235 | * |
| 7.77 | Camphene | 3.00 | 1.18 | 4.92 | 1.37 | 0.1404 | ns |
| 8.98 | Hexanal | 13.68 | 3.14 | 4.44 | 0.37 | 0.0072 | ** |
| 9.45 | Methyl valerate | 0.00 | 0.00 | 2.03 | 0.87 | 0.0155 | * |
| 10.86 | Unknown_2 | 3.09 | 3.64 | 1.20 | 0.74 | 0.4294 | ns |
| 11.07 | Unknown_3 | 4.44 | 5.04 | 1.75 | 0.82 | 0.4131 | ns |
| 11.4 | *o*-Xylene | 0.84 | 0.37 | 0.89 | 0.11 | 0.8529 | ns |
| 11.72 | *p*-Xylene | 0.85 | 0.31 | 1.20 | 0.26 | 0.2145 | ns |
| 12.98 | α-Phellandrene | 7.37 | 1.42 | 15.24 | 2.17 | 0.0063 | ** |
| 13.71 | *m*-Xylene | 0.94 | 0.24 | 1.01 | 0.18 | 0.7035 | ns |
| 14.1 | 2-Heptanone | 3.99 | 1.28 | 1.78 | 0.17 | 0.0422 | * |
| 14.3 | Unknown_4 | 2.05 | 0.54 | 9.05 | 3.84 | 0.0354 | * |
| 14.38 | Caproic acid methyl ester | 5.64 | 1.61 | 13.76 | 5.32 | 0.0647 | ns |
| 14.52 | Limonene | 5.63 | 1.16 | 11.15 | 2.69 | 0.0309 | * |
| 14.8 | Terpene 1 | 0.29 | 0.05 | 0.41 | 0.13 | 0.2245 | ns |
| 15.22 | 2-Hexenal | 0.49 | 0.22 | 1.07 | 0.06 | 0.012 | * |
| 15.32 | Dodecane | 0.45 | 0.36 | 0.00 | 0.00 | 0.0957 | ns |
| 15.61 | Unknown_5 | 0.83 | 0.24 | 0.00 | 0.00 | 0.004 | ** |
| 16.19 | Unknown_6 | 0.76 | 0.20 | 0.00 | 0.00 | 0.0029 | ** |
| 16.23 | Unknown_7 | 0.33 | 0.04 | 0.00 | 0.00 | 0.0002 | ** |
| 17.26 | Cymene | 5.09 | 0.84 | 14.55 | 1.56 | 0.0008 | ** |
| 18.08 | Methyl heptanoate | 4.06 | 1.03 | 9.18 | 3.90 | 0.0929 | ns |
| 19.44 | 6-Methyl-5-hepten-2-one | 2.35 | 0.69 | 2.07 | 0.28 | 0.5473 | ns |
| 20.54 | Unknown_9 | 7.79 | 2.76 | 1.69 | 0.16 | 0.0188 | * |
| 20.98 | Caprylic acid methyl ester | 2.49 | 0.49 | 7.29 | 2.34 | 0.0254 | * |
| 21.69 | Sesquiterpene_1 | 1.60 | 0.46 | 0.66 | 0.16 | 0.0291 | ** |
| 22.23 | Acetic acid | 2.45 | 0.47 | 6.80 | 0.83 | 0.0014 | ** |
| 22.58 | Sesquiterpene_2 | 2.12 | 0.53 | 1.01 | 0.16 | 0.0262 | * |
| 22.73 | Sesquiterpene_3 | 4.13 | 0.74 | 4.62 | 5.34 | 0.8821 | ns |
| 22.94 | Unknown_10 | 0.31 | 0.06 | 0.00 | 0.00 | 0.0009 | ** |
| 23.34 | Sesquiterpene_4 | 1.06 | 0.09 | 0.23 | 0.06 | 0.0002 | ** |
| 23.47 | Nonanoic acid methyl ester | 0.56 | 0.11 | 0.55 | 0.21 | 0.9359 | ns |
| 23.54 | Unknown_11 | 0.55 | 0.07 | 0.29 | 0.04 | 0.0045 | ** |
| 23.85 | Benzaldehyde | 2.21 | 0.28 | 3.13 | 0.48 | 0.0439 | * |
| 23.95 | Sesquiterpene_5 | 2.23 | 0.30 | 0.75 | 0.14 | 0.0015 | ** |
| 24.25 | Sesquiterpene_6 | 5.43 | 0.85 | 2.78 | 0.48 | 0.0095 | ** |
| 24.41 | Sesquiterpene_7 | 2.24 | 0.47 | 0.80 | 0.17 | 0.0075 | ** |
| 24.68 | β-Linalool | 1.04 | 0.12 | 1.04 | 0.13 | 0.9967 | ns |
| 24.93 | 2-Methyl propanoic acid | 2.47 | 0.59 | 9.21 | 0.74 | 0.0002 | ** |
| 25.12 | Sesquiterpene_8 | 5.13 | 0.89 | 2.20 | 0.18 | 0.0051 | ** |
| 25.26 | Bornyl acetate | 2.27 | 0.18 | 1.50 | 0.27 | 0.0151 | * |
| 25.49 | α-Bergamotene | 18.15 | 2.15 | 11.23 | 1.80 | 0.0129 | * |
| 25.61 | Caryophyllene | 70.10 | 8.85 | 59.60 | 3.87 | 0.1327 | ns |
| 26.54 | Borneol butyrate | 4.95 | 0.70 | 9.94 | 2.08 | 0.0171 | * |
| 26.67 | Isocaryophyllene | 1.42 | 0.15 | 1.19 | 0.16 | 0.1362 | ns |
| 26.96 | Sesquiterpene_9 | 0.80 | 0.11 | 0.00 | 0.00 | 0.0002 | ** |
| 27.07 | Humulene | 11.25 | 1.03 | 10.61 | 0.76 | 0.434 | ns |
| 27.14 | Sesquiterpene_10 | 1.53 | 0.35 | 1.27 | 0.10 | 0.2818 | ns |
| 27.31 | Unknown_12 | 3.77 | 0.50 | 3.42 | 0.45 | 0.4215 | ns |
| 27.45 | *trans*-β-Farnesene | 3.49 | 0.29 | 2.52 | 0.57 | 0.0604 | ns |
| 27.52 | Υ-muurolene | 3.87 | 0.31 | 1.31 | 0.34 | 0.0006 | ** |
| 27.86 | Germacrene D | 26.40 | 2.82 | 10.18 | 1.53 | 0.0009 | ** |
| 27.98 | Sesquiterpene_11 | 2.81 | 0.36 | 1.18 | 0.31 | 0.0042 | ** |
| 28.06 | Sesquiterpene_12 | 1.09 | 0.41 | 0.91 | 0.14 | 0.5165 | ns |
| 28.13 | Sesquiterpene_13 | 1.85 | 0.33 | 0.40 | 0.16 | 0.0023 | ** |
| 28.21 | Sesquiterpene_14 | 1.39 | 0.35 | 0.75 | 0.17 | 0.0464 | ** |
| 28.33 | Unknown_13 | 1.30 | 0.13 | 2.14 | 0.37 | 0.0205 | * |
| 28.85 | δ-cadinene | 13.13 | 1.10 | 4.40 | 1.10 | 0.0006 | ** |
| 29.09 | β-Sesquiphellandrene | 0.20 | 0.03 | 0.33 | 0.05 | 0.0165 | * |
| 29.28 | Sesquiterpene_15 | 0.52 | 0.02 | 0.19 | 0.04 | 0.0003 | ** |
| 29.48 | Sesquiterpene_16 | 1.40 | 0.28 | 2.26 | 2.73 | 0.6146 | ns |
| 29.66 | Benzenic ring | 0.68 | 0.09 | 0.24 | 0.04 | 0.0014 | ** |

**Table 2.** *Cont.*

| RT [a] | Compounds | Wild Arnica | | Cultivated Arnica | | | |
|---|---|---|---|---|---|---|---|
| | | Mean [b] | ±SD [c] | Mean [b] | ±SD [c] | *p*-Value | Signif. Code |
| 30.16 | Calamenene | 3.38 | 1.09 | 3.47 | 0.74 | 0.9112 | ns |
| 30.33 | Caproic acid | 1.67 | 0.24 | 2.15 | 0.21 | 0.0623 | ns |
| 30.87 | Unknown_14 | 2.12 | 0.25 | 1.34 | 0.48 | 0.0668 | ns |
| 32.15 | Unknown_15 | 1.24 | 0.14 | 0.68 | 0.12 | 0.0068 | ** |
| 32.31 | Caryophyllene epoxide | 0.58 | 0.06 | 0.86 | 0.19 | 0.0706 | ns |
| 34.13 | Unknown_16 (phenolic compound) | 0.26 | 0.04 | 0.63 | 0.09 | 0.0028 | ** |
| 34.31 | Phenol, 5-methyl-2-(1-methylethyl) | 0.20 | 0.05 | 0.51 | 0.10 | 0.0083 | ** |

RT [a]: retention time (min); Mean [b]: Mean value (*n* = 5); Data are expressed in µg/g; SD [c]: Standard deviation (*n* = 5); Signif. Code [d]: *, *p* < 0.05; **, *p* < 0.01; ns, not significant.

## 4. Discussion

Looking at the results of the germination trials, the germination percentage (GRP) is high (>75%) both for cultivated and wild seeds, regardless of the treatment. This means that both for the cultivated arnica and the wild one harvested in Valsaviore, it is possible to obtain a good germination rate without the use of phytohormones, that are rather costly. Cultivated seeds differ from the wild ones mainly because they germinated faster, no matter what the hormonal treatment was or if they were in the control plot. This can be considered an irrelevant aspect as the germination peak for cultivated seeds was reached only a few days (from two to four days) before the wild ones. The faster germination of cultivated seeds is expected since, differently from wild arnica, they were subjected to a certain selection during the propagation, with the slower and less vital seeds discarded during the production procedure or at the quality control by the seedlings company. Also, the more evident effect on wild seeds germination speed (GSP) of GA3 treatments (faster with increasing concentration of GA in the solution) could be due to the above consideration (cultivated seeds are more uniform and they germinate in an easier way, so they do not need stimulating hormones such as GA3). Despite the acceptably high GRP and GSP, further trials to improve them could be some treatments with laccase to increase the rate of seed coat degradation [42,43] and treatment with sulfuric acid followed by manual cleaning [44], which apparently increased the germination of orchid and wild grasses seeds, respectively, and could also be tested on alpine species such as arnica. Low temperatures could also favor the germination of alpine plant seed as found by [45]. The fact that the spontaneous arnica from Valsaviore has no particular issues during germination could boost the creation of local nurseries in the mountains to farm native germplasm and contribute to conservation actions as well as entrepreneurial activities in marginal areas. As the farm renew the crops purchasing the seedlings every three years, the self-production of seedlings would importantly reduce the expenses for consumables and make this activity in marginal territories more sustainable and environmentally friendly. The native germplasm, then, could be employed not only for the herbal ointments production but also for nature conservation actions as environmental remediation of alpine pasture habitats and/or restocking interventions. An interesting hypothesis was suggested in [46]: the possibility of using the rosettes coming from genets produced by the exhausted plantation as propagation material. The Valsaviore farm referred that an exhausted field of arnica completely colonized a great part of the grassland after being left untouched for ten years. Even if what is referred was just an anecdotal case, it could be an interesting hypothesis from a conservation point of view. In mountain territories, were there is usually an abundance of land, it could be sustainable to consider some years of unproductive plantation to give *A. montana* the time for a natural regeneration, and then obtain a restored and sustainably productive habitat. The restoration of natural-like habitats, to be used as basis for conservation and herbal plants production, would be advisable.

Further studies should be implemented also in the phytochemical characterization aspect. According to HPLC and NMR analysis, wild arnica resulted significantly richer in two unknown compounds, not recognizable due to the complexity of the matrix. Cultivated

arnica resulted richer in the three identified compounds HM (6-O-methacryloylhelenalin), HMB (6-O-(2-methylbutyryl)-helenalin) and HIB (6-O-(2-methylbutyryl)-helenalin). HMB and HM exhibited a stronger activity in the NF-κB EMSA and IL-8 ELISA in vitro assays. Sesquiterpene lactones seem to be the most important NF-κB inhibiting compounds in the arnica extract [47] similarly to corticoid steroids [14], and these compounds were found both in the cultivated and wild accessions from Valsaviore. Other chromatographic techniques (such as HPLC-MS) could be useful to enhance arnica phytochemical characterization. Also, some biological tests would clarify the differences among the two arnica genotypes. Differently from other MAPs (medicinal and aromatic plants), arnica was not subjected to a deep and systematic selection [19,48], for what concerns the bioactive compounds content. Till today, the quality control of arnica is estimated as total amount of sesquiterpene lactones content expressed as equivalents of dihydrohelenalin tiglate, without considering the different compounds, and by considering the yield in extracts or essential oils. From our study, the presence of sesquiterpene compounds like the esters of helenaline (H) and dehydroelenaline (DH) was confirmed in both the cultivated and wild arnica, and the NMR and HPLC chromatograms resulted very similar (Figures 3 and 4) and the extract yield was comparable. Wild and cultivated arnica also contained almost all the same compounds in the volatiles fraction, that differed only quantitatively for 44 out of 71 compounds identified, so we can conclude that the wild arnica ecotype characterized in the study is suitable for herbal use.

Looking further at the volatile composition, a significantly high percentage of acetic acid methyl ester (38 μg/g) and the 2-methyl ester of propanoic acid (31 μg/g) were found for cultivated arnica through SPME-GC-MS analysis, rather than other straight-chain and branched hydrocarbons. An increase in the quantity of acetic acid and acetic acid methyl ester has been associated with fermentation processes [49]. As referred by the farmers, cultivated arnica was dried using the traditional method of air drying on trellis protected from light and oven dried for the leftover material when trellises were full, so if the cultivated arnica flos were immediately dried after the harvesting, we cannot exclude some fermentation processes due to the use of the traditional method, more subject to environmental factors (such as air humidity or temperature changes) or an excessive layering in the oven or on the trellises. The presence of compounds associated to fermentation processes was found also in commercial arnica purchased from the web (unpublished data), because this drying method is very common in marginal territories (Central and Eastern Europe) where arnica is collected/cultivated. Further studies should be done on the most effective drying conditions, in particular temperature and moisture and drying time should be monitored, to increase the value and quality of local products. Traditional drying methods are often still used in the mountains and a better training of farmers on the quality control of the product should be implemented. The quality analysis of officinal and aromatics plants is fundamental for mountain and marginal territories small enterprises to obtain quality products from a perspective of multifunctionality [50,51]. The investigation of the bioactive compounds content in other parts of the plant such as leaves, rhizomes, roots, and seeds could provide moreover guidelines for a possible broader utilization of the plant resources [52].

The differences in the other volatile composition should instead be attributed to other edaphic or biological conditions, such as the different germplasm or soil. Age of the plant, season, microbial attacks, grazing, radiation, competition, and nutritional status have been proven to have an impact on the secondary metabolite profile in higher plants, that is then influenced then by many and different factors [53,54]. Although the wild arnica was not cultivated in exactly the same field, the climate in the area was the same and it has been already demonstrated that the total contents of sesquiterpene lactones and flavonoids were not positively correlated with the altitude of the growing site for *A. montana* in the Tyrolean Alps [47]. Monoterpenes, sesquiterpene and other fragrance compounds are commonly considered to have an antifeedant activity; none of the known biological activities of sesquiterpenoids is in any way related to factors, which are changing

with the altitude of the growing site, e.g., sesquiterpenoids neither absorb radiation in the UV-B range nor do they have pronounced radical scavenging activity [55]. An influence of germplasm on the differences in the secondary metabolite profile of the considered ecotypes cannot be ruled out based on the present data and although it is of minimal importance for the purposes of the study, that sought to explore the eligibility of wild arnica for herbal production, which was verified. Further studies need to be realized, and the ultimate proof for a genetic basis of the observed variation will require: (a) genetic characterization (b) cultivation experiments under identical growing conditions, that would also give cues for the productive performance after the germination performance (quantity of inflorescences per plant) as done by the authors for an ancient cultivar of maize in the Alps [56]. Further, drying the flower-heads from plants grown in the same conditions with a defined drying protocol would provide better knowledge of the secondary metabolism of *A. montana* plants excluding the impact of drying conditions. This again testifies to the importance of the first transformation of arnica flower-heads on their quality. However, as recently demonstrated for related New Zealand neophytes from the Lactuceae tribe of the Asteraceae family, phytochemical differences between different populations are usually more pronounced than intraspecific variations of populations from a particular taxon [48].

In the cultivated arnica a higher quantity of α-phellandrene, β-sesquiphellandrene, limonene, cymene, bornyl acetate and borneol butyrate was found. α-Phellandrene and limonene are monoterpenes commonly used in fragrances and their anti-inflammatory properties have been widely tested [57–59]. Bornyl acetate and borneol butyrate are the acetate ester and the butyrate of borneol, a bicyclic organic compound and a terpene derivative with demonstrated anti-inflammatory properties [60]. In the wild arnica, a higher amount of hexanal was detected, while in cultivated arnica was found a corresponding amount of 2-hexanal was detected. Hexanal and benzaldehyde are naturally occurring antimicrobial aldehydes [61]. α-Bergamotene, found in a higher quantitative in wild arnica, is very common in citrus and has proven antimicrobial and other important biological properties [62]. Germacrene D is an important precursor of sesquiterpenes [63] and has been already mentioned as one of the principal components of the secondary metabolite profile in arnica essential oil [10]; this last compound was found in a higher quantity in the wild flos together with the sesquiterpene Y-muurolene and some other compounds identified as sesquiterpenes based on their mass fragmentation patterns.

## 5. Conclusions

In this study we compared wild and cultivated arnica showing that the wild germplasm is suitable for farming for herbal use (based on both the chemical composition and germination performance) while preserving local populations. In applied botany, the possibility to grow *A. montana* in marginal territories is of interest for farmers in alpine regions looking for alternative high mountain farming crops, especially in regions like Valsaviore. The drying conditions are important to obtain a quality product, since this could affect the volatiles composition of arnica flos used to produce herbal ointments. A controlled local first transformation is important to incentivize the local production and to provide industry with a good raw material and to have a local value-add. The production of seedlings "in loco" could also be of great interest for farmers (who often renew the crops with external propagation material) and for natural conservation purposes. As we saw from our study, wild and cultivated arnica seeds do not present very different germination performances and further study should be performed to support small and medium enterprises for the local production of seedlings. Also, if lacking a genetic characterization, the phytochemical variations in the two populations studied are not totally attributable to different germplasm, it is worth mentioning that very little agronomic selection was done on *A. montana*, even on the most recognized commercial cultivar "Arbo". Just one comparative study on Spanish arnica populations was done in 2009 [64], to support the cultivation of *A. montana* in New Zealand, and one promising cultivar producing more active metabolites was individuated without, though, a following experimental research under identical growing conditions.

Genetic adaptation to the specific environment of the growing site is an important factor to be considered. Studies assessing differences between wild populations of *A. montana* are underway and could give further cues on the different contributions of genetic adaption versus environmental plasticity for the observed phytochemical differences between arnica plants [48]. These studies are particularly needed considering the low genetic improvement and the increasing necessity of herbal products coming from cultivated material to protect and preserve the wild population from excessive collection pressure.

**Author Contributions:** Conceptualization, V.L., L.G., A.B., G.B. and A.G.; methodology, V.L., L.G. and G.B.; software, L.G.; validation, A.B.; formal analysis, V.L., A.R., G.B., L.G., M.Z. and D.P.; investigation: V.L. and G.B.; data curation, V.L., L.G. and D.P.; writing—original draft preparation: L.G., V.L. and G.B.; writing—review and editing, V.L., M.Z. and A.R.; supervision, A.G. and A.B.; project administration, A.G.; funding acquisition, A.G. All authors have read and agreed to the published version of the manuscript.

**Funding:** This research was supported by the Department for Regional Affairs and Autonomies (DARA) of the Italian Presidency of the Council of Ministers (DARA-CRC Ge.S.Di.Mont. agreement), FISR-MIUR "Italian Mountain Lab" project, and by the "Montagne: Living Labs di innovazione per la transizione ecologica e digitale" project.

**Institutional Review Board Statement:** Not applicable.

**Informed Consent Statement:** Not applicable.

**Data Availability Statement:** Not applicable.

**Acknowledgments:** We wish to thank Shanty Mae farm (Saviore-BS) that have contributed to this study. This research was supported by "Italian Mountain Lab" project, "DARA—CRC Ge.S.Di.Mont." agreement and and by "Montagne: Living Labs di innovazione per la transizione ecologica e digitale" project.

**Conflicts of Interest:** The authors declare no conflict of interest. The funders had no role in the design of the study; in the collection, analyses, or interpretation of data; in the writing of the manuscript, or in the decision to publish the results.

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
