# Peer review of "Comparing Wild and Cultivated Arnica montana L. from the Italian Alps to Explore the Possibility of Sustainable Production Using Local Seeds"

_sustainability, doi:10.3390/su13063382_

Round 1

Reviewer 1 Report

This MS presents the results of a germination performance and phytochemical content comparison between wild, cultivated, and commercial accessions of Arnica montana in Valsaviore (Italian Alps). The aim of the study was to provide evidence for the adequacy of wild germplasm of A. montana to support local cultivation and production of arnica related herbal medicinal products. This is a quite important issue, considering the collection pressure on natural populations of A. montana caused by herbal industry. The scope of this MS is valuable as it combines nature conservation with sustainable local employment and entrepreneurship.

The study uses a simple, non-destructive, and economical technique for sample preparation and an evolving technique for the biologically active compounds determination that are well-presented. Germination trials are also well-presented in the Methodology.

However, there are several issues regarding the publication suitability of this MS in Sustainability journal:

  1. For the phytochemical characterization, wild accessions are compared to cultivated ones but for the germination trials to commercial accessions. It is stated that the commercial seeds were purchased from the nursery that provides the seedlings to the cultivation farm. But, considering the effect of genetic and phenotypic adaptation to the environment to the phytochemical composition of plants these results cannot be comparable. Collecting seeds also from the cultivated Arnica and picking flower-heads also from the nursery would have increased the power of experimental design.
  2. For the phytochemical characterization different drying methods were used for the wild flos (immediate transportation and oven drying, at 25-30 °C for one week) and the cultivated arnica (air drying on trellis). Again, this makes results not comparable since drying is one of the most essential processing steps (L. 99) as it could affect dramatically the volatiles composition (L. 482-483) due to fermentation processes.
  3. Results of phytochemical analyses are not presented in a clear, direct way that would allow the reader to decide on the competence of wild flos use for local cultivation. Numerous compounds are mentioned, but there is no info on which of them are the most important in arnica based herbal products or the most promising for future studies. The MDS bi-plot illustrates that the two (although, actually three) accessions are covering different parts of space, but there is a lack on the actual meaning of these parts. Indeed, this statistical analysis, although very crucial for the MS hypothesis, is “hidden” and not well-presented.
  4. It would be interesting to see similar phytochemical comparisons not only for flower-heads but also for other plant parts such as leaves, rhizomes, roots, and seeds (e.g., 1111/jphp.12724) in order to give guidelines for a possible broader utilization of the plant resources.
  5. In the same way different propagation techniques, such as with clone seedlings or rosettes (e.g., 1155/2013/414363) would have enriched the knowledge on the potential of local cultivation.

Author Response

Comments and Suggestions for Authors_1

This MS presents the results of a germination performance and phytochemical content comparison between wild, cultivated, and commercial accessions of Arnica montana in Valsaviore (Italian Alps). The aim of the study was to provide evidence for the adequacy of wild germplasm of A. montana to support local cultivation and production of arnica related herbal medicinal products. This is a quite important issue, considering the collection pressure on natural populations of A. montana caused by herbal industry. The scope of this MS is valuable as it combines nature conservation with sustainable local employment and entrepreneurship.

The study uses a simple, non-destructive, and economical technique for sample preparation and an evolving technique for the biologically active compounds determination that are well-presented. Germination trials are also well-presented in the Methodology.

Thank you for your positive comment on the scope of our study combining natural conservation and local sustainable development of mountain agricultural activities. We examined all the manuscript following your revision and checked English grammar and spelling.

However, there are several issues regarding the publication suitability of this MS in Sustainability journal:

For the phytochemical characterization, wild accessions are compared to cultivated ones but for the germination trials to commercial accessions. It is stated that the commercial seeds were purchased from the nursery that provides the seedlings to the cultivation farm. But, considering the effect of genetic and phenotypic adaptation to the environment to the phytochemical composition of plants these results cannot be comparable. Collecting seeds also from the cultivated Arnica and picking flower-heads also from the nursery would have increased the power of experimental design.

Thank you for your comment. We understand that in the article this aspect is not clearly explained, and then the experimental design could appear inadequate. The farm does not use its own seeds and renew every three years the crops with the seedling purchased from the mentioned nursery. So, the arnica plants cultivated by Shanty Mae farms comes exclusively from the nursery. We explained it better in the article (lines 137-139). We also changed all the “commercial” in cultivated to avoid this kind of ambiguities (highlighted in red along the article). The local production of seedlings would improve the economical and environmental sustainability and it is the main reason why the farm proposed us this experimentation, as we added in lines 389-392.

We totally agree that the effect of genetic and phenotypic adaptation is to investigate, as we mentioned in the article (in the conclusion, lines 516-522) and also comparing the flower-heads from Valle d’Aosta where the nursery is located (same seeds/seedlings – different area) would have been interesting but it was not the main target of our study, that was focused on the production in Valsaviore.

For the phytochemical characterization different drying methods were used for the wild flos (immediate transportation and oven drying, at 25-30 °C for one week) and the cultivated arnica (air drying on trellis). Again, this makes results not comparable since drying is one of the most essential processing steps (L. 99) as it could affect dramatically the volatiles composition (L. 482-483) due to fermentation processes.

The aim of the study was also to evaluate the sustainability of a local production, so we retained interesting to characterize the inflorescences locally dried. As we say at lines 27-28, “the possibility to grow A. montana and a controlled local first transformation are important to incentivize a local, good quality and sustainable production”.  Essentially, the scope of the study was achieved, and differences are of minor importance (we specified this at lines 463-466). We understand this can affect the experimental design, but it can give also cues on the necessity of formulating appropriate drying protocols (lines 89-91). This is further significant considering that we found the presence of compounds associated to fermentation processes in commercial arnica purchased in the web (unpublished data), because this drying method is very common in marginal territories (Central and Eastern Europe) where arnica is collected/cultivated, as we said at lines 438-441. Anyway, as we agree with you on this aspect, we added a phrase at lines 471-475, where we expose the limits of this study and the necessity of further investigations also from the genetic and productive point of view. This is also clear in the title, where it is written that this study is to “explore” the possibility of a local production of seedlings using local germplasm. A study on arnica chemovars without considering identical growing condition was considered anyway useful to increase knowledge on arnica secondary metabolites (61: Perry, N.B.; Burgess, E.J.; Rodríguez Guitián, M.A.; Romero Franco, R.; López Mosquera, E.; Smallfield, B.M.; Joyce, N.I.; Littlejohn, R.P. Sesquiterpene Lactones in Arnica montana: Helenalin and Dihydrohelenalin Chemotypes in Spain. Planta Med. 2009, 75, 660-666.)

Results of phytochemical analyses are not presented in a clear, direct way that would allow the reader to decide on the competence of wild flos use for local cultivation. Numerous compounds are mentioned, but there is no info on which of them are the most important in arnica based herbal products or the most promising for future studies. The MDS biplot illustrates that the two (although, actually three) accessions are covering different parts of space, but there is a lack on the actual meaning of these parts. Indeed, this statistical analysis, although very crucial for the MS hypothesis, is “hidden” and not well-presented.

As we say in the article, “differently from other MAPs (medicinal and aromatic plants), arnica was not subjected to a deep and systematic selection [19-47], for what concerns the bioactive compounds content”. (lines 416-418). From our knowledge, in the quality control of arnica is considered the total amount of sesquiterpene lactones expressed as equivalent of Dihydrohelenalin Tiglate, without considering the different compounds. This is then just an estimation of the bioactive compounds content and our study is a more in-depth characterization, both for volatiles and lactones sesquiterpenes. The NMR profile is very similar and targeted towards sesquiterpenes lactones, so this is the most important parameter, in our opinion, to say that also the wild chemotype is eligible for herbal production (see results explained at lines 277-278 and 300-302). We tried to make this clearer and explained in a more directed way in the article as asked (lines 421-427).

 Volatiles compounds important for Arnica quality for us are quite well described in the discussion (lines 479-493), we added some details on sesquiterpenes lactones (lines 409-413). We added more details on MDS analysis in materials and methods (lines 250-255). We tried to present better the results of the MDS analysis and improve the caption of figure 6.

It would be interesting to see similar phytochemical comparisons not only for flower-heads but also for other plant parts such as leaves rhizomes, roots, and seeds (e.g., 1111/jphp.12724) in order to give guidelines for a possible broader utilization of the plant resources.

We agree it would be interesting. However, our aim was to investigate the part of the flower traditionally used. We suggest also this work from Douglas et al. (2004): “Sesquiterpene Lactones in Arnica montana: a Rapid Analytical Method and the Effects of Flower Maturity and Simulated Mechanical Harvesting on Quality and Yield”, that consider different flower parts and in particular shows the minor content of sesquiterpene lactones in the stem of the flower. We added the reference suggested in the paper as a possible further investigation.

In the same way different propagation techniques, such as with clone seedlings or rosettes (e.g., 1155/2013/414363) would have enriched the knowledge on the potential of local cultivation.

Thank you for your suggestion. We mentioned this work (line 395) of Sugier et al. (2013): “Propagation and Introduction of Arnica montana L. into Cultivation: A Step to Reduce the Pressure on Endangered and High-Valued Medicinal Plant Species.” The farm referred that they tried to maintain an exhausted field, that after about ten years completely colonized the grassland. We described this in the article as an interesting case (397-403). 

Reviewer 2 Report

Dear Authors,

The manuscript entitled: "Comparing wild and cultivated Arnica montana L. from Italian Alps to explore the possibility of a sustainable production using local seeds" can be an example of a good manuscript, evaluating several phytochemical parameters and seeds germination comparison of the plant samples. In general I think some parts (e.g. Introduction, some parts of Methods like plant material) is too long and can be somehow tedious for the readers, therefore please make them shorter but concise. Please find my comments below for improving the quality of manuscript:

Abstract

L24: please revise "significantly" to "significant"

- I would prefer to combine the keywords: "alpine plants and medicinal plants"

Introduction

- In my opinion this part is too long, it may better to shorten

L127: please correct "Basing on these considerations" to "Based on ..."

Materials and methods

- please revise "ml" to "mL" through whole text, as well as "l" to "L"

L223: please correct: "Was used a linear gradient from 55:45 to 40:60 water/methanol over 35 min. [16]."

L 230: According to authors: "Fraction f2 and f4 corresponded to 230
the Sesquiterpene lactones", please describe how did you distinguish?

Results

- In my opinion the figures and tables should be brought after section which is mentioned for the first time, following up the readers can be much easier in this case, I do not recommend to use a separate section as figures and tables

- Please replace Figure 5 with a better quality one

- In case of Table 2., decreasing font size should be done

Conclusion

- Please make "A. montana" italic in whole parts

References

- Please check one more this section, in case of being italic (specifically plants' names e.g. L531,545,549: Arnica montana), boldness, etc.

Good luck!

Author Response

Comments and Suggestions for Authors_2

Dear Authors,

The manuscript entitled: "Comparing wild and cultivated Arnica montana L. from Italian Alps to explore the possibility of a sustainable production using local seeds" can be an example of a good manuscript, evaluating several phytochemical parameters and seeds germination comparison of the plant samples. In general I think some parts (e.g. Introduction, some parts of Methods like plant material) is too long and can be somehow tedious for the readers, therefore please make them shorter but concise. Please find my comments below for improving the quality of manuscript:

Thank you for your positive opinion and your suggestions to improve our manuscript. However, the multidisciplinary character of the article put us a bit in difficult to reduce too much the framework in the introduction and an accurate description of Material and Method was in our opinion necessary to ensure the replicability of the study and a correct interpretation of the results. We tried in any case to shorten the mentioned parts as much as possible. We examined all the manuscript following your revision and checked English grammar and spelling.

Abstract

L24: please revise "significantly" to "significant"

Corrected.

- I would prefer to combine the keywords: "alpine plants and medicinal plants"

 We agree and combine the keywords in “alpine medicinal plants”.

Introduction

- In my opinion this part is too long, it may better to shorten

We tried to shorten the introduction as much as possible.

L127: please correct "Basing on these considerations" to "Based on ..."

 Corrected, thank you.

Materials and methods

- please revise "ml" to "mL" through whole text, as well as "l" to "L"

We revised all the text, thank you for your amendment.

L223: please correct: "Was used a linear gradient from 55:45 to 40:60 water/methanol over 35 min. [16]."

Corrected.

L 230: According to authors: "Fraction f2 and f4 corresponded to 230

the Sesquiterpene lactones", please describe how did you distinguish?

We explained better the separation and chemical characterization of the fractions of arnica extract in lines 203-211.

Results

- In my opinion the figures and tables should be brought after section which is mentioned for the first time, following up the readers can be much easier in this case, I do not recommend to use a separate section as figures and tables

We agree on what you say, but we had to follow the rules of the paper template.

- Please replace Figure 5 with a better quality one

We improved figure 5.

- In case of Table 2., decreasing font size should be done

 Thank you for your advice. We improved the table. 

Conclusion

- Please make "A. montana" italic in whole parts

 Done

References

- Please check one more this section, in case of being italic (specifically plants' names e.g. L531,545,549: Arnica montana), boldness, etc.

 We checked this section.

Good luck!

Thank you for your help!

Round 2

Reviewer 1 Report

Dear Authors,

Thank you for your efforts to respond to my comments. I suggest that this MS is suitable for publication now.

Please, although it is improved, consider to re-edit English language, for example avoid long sentences in the text.

All the best